# A Baseline for Detecting Misclassified and Out-of-Distribution Examples in Neural Networks

**Dan Hendrycks**[*]
University of Chicago
dan@ttic.edu

**Kevin Gimpel**
Toyota Technological Institute at Chicago
kgimpel@ttic.edu

## Abstract

We consider the two related problems of detecting if an example is misclassified or out-of-distribution. We present a simple baseline that utilizes probabilities from softmax distributions. Correctly classified examples tend to have greater maximum softmax probabilities than erroneously classified and out-of-distribution examples, allowing for their detection. We assess performance by defining several tasks in computer vision, natural language processing, and automatic speech recognition, showing the effectiveness of this baseline across all. We then show the baseline can sometimes be surpassed, demonstrating the room for future research on these underexplored detection tasks.

## 1 Introduction

When machine learning classifiers are employed in real-world tasks, they tend to fail when the training and test distributions differ. Worse, these classifiers often fail silently by providing high-confidence predictions while being woefully incorrect (Goodfellow et al., 2015; Amodei et al., 2016). Classifiers failing to indicate when they are likely mistaken can limit their adoption or cause serious accidents. For example, a medical diagnosis model may consistently classify with high confidence, even while it should flag difficult examples for human intervention. The resulting unflagged, erroneous diagnoses could blockade future machine learning technologies in medicine. More generally and importantly, estimating when a model is in error is of great concern to AI Safety (Amodei et al., 2016).

These high-confidence predictions are frequently produced by softmaxes because softmax probabilities are computed with the fast-growing exponential function. Thus minor additions to the softmax inputs, i.e. the logits, can lead to substantial changes in the output distribution. Since the softmax function is a smooth approximation of an indicator function, it is uncommon to see a uniform distribution outputted for out-of-distribution examples. Indeed, random Gaussian noise fed into an MNIST image classifier gives a "prediction confidence" or predicted class probability of 91%, as we show later. Throughout our experiments we establish that the prediction probability from a softmax distribution has a poor direct correspondence to confidence. This is consistent with a great deal of anecdotal evidence from researchers (Nguyen & O'Connor, 2015; Yu et al., 2010; Provost et al., 1998; Nguyen et al., 2015).

However, in this work we also show the prediction probability of incorrect and out-of-distribution examples tends to be lower than the prediction probability for correct examples. Therefore, capturing prediction probability statistics about correct or in-sample examples is often sufficient for detecting whether an example is in error or abnormal, even though the prediction probability viewed in isolation can be misleading.

These prediction probabilities form our detection baseline, and we demonstrate its efficacy through various computer vision, natural language processing, and automatic speech recognition tasks. While these prediction probabilities create a consistently useful baseline, at times they are less effective, revealing room for improvement. To give ideas for future detection research, we contribute

---

[*]Work done while the author was at TTIC. Code available at github.com/hendrycks/error-detection

one method which outperforms the baseline on some (but not all) tasks. This new method evaluates the quality of a neural network's input reconstruction to determine if an example is abnormal.

In addition to the baseline methods, another contribution of this work is the designation of standard tasks and evaluation metrics for assessing the automatic detection of errors and out-of-distribution examples. We use a large number of well-studied tasks across three research areas, using standard neural network architectures that perform well on them. For out-of-distribution detection, we provide ways to supply the out-of-distribution examples at test time like using images from different datasets and realistically distorting inputs. We hope that other researchers will pursue these tasks in future work and surpass the performance of our baselines.

In summary, while softmax classifier probabilities are not directly useful as confidence estimates, estimating model confidence is not as bleak as previously believed. Simple statistics derived from softmax distributions provide a surprisingly effective way to determine whether an example is misclassified or from a different distribution from the training data, as demonstrated by our experimental results spanning computer vision, natural language processing, and speech recognition tasks. This creates a strong baseline for detecting errors and out-of-distribution examples which we hope future research surpasses.

## 2  PROBLEM FORMULATION AND EVALUATION

In this paper, we are interested in two related problems. The first is **error and success prediction**: can we predict whether a trained classifier will make an error on a particular held-out test example; can we predict if it will correctly classify said example? The second is **in- and out-of-distribution detection**: can we predict whether a test example is from a different distribution from the training data; can we predict if it is from within the same distribution?[1] Below we present a simple baseline for solving these two problems. To evaluate our solution, we use two evaluation metrics.

Before mentioning the two evaluation metrics, we first note that comparing detectors is not as straightforward as using accuracy. For detection we have two classes, and the detector outputs a score for both the positive and negative class. If the negative class is far more likely than the positive class, a model may always guess the negative class and obtain high accuracy, which can be misleading (Provost et al., 1998). We must then specify a score threshold so that some positive examples are classified correctly, but this depends upon the trade-off between false negatives (fn) and false positives (fp).

Faced with this issue, we employ the Area Under the Receiver Operating Characteristic curve (AUROC) metric, which is a threshold-independent performance evaluation (Davis & Goadrich, 2006). The ROC curve is a graph showing the true positive rate ($\text{tpr} = \text{tp}/(\text{tp} + \text{fn})$) and the false positive rate ($\text{fpr} = \text{fp}/(\text{fp} + \text{tn})$) against each other. Moreover, the AUROC can be interpreted as the probability that a positive example has a greater detector score/value than a negative example (Fawcett, 2005). Consequently, a random positive example detector corresponds to a 50% AUROC, and a "perfect" classifier corresponds to 100%.[2]

The AUROC sidesteps the issue of threshold selection, as does the Area Under the Precision-Recall curve (AUPR) which is sometimes deemed more informative (Manning & Schütze, 1999). This is because the AUROC is not ideal when the positive class and negative class have greatly differing base rates, and the AUPR adjusts for these different positive and negative base rates. For this reason, the AUPR is our second evaluation metric. The PR curve plots the precision ($\text{tp}/(\text{tp} + \text{fp})$) and recall ($\text{tp}/(\text{tp} + \text{fn})$) against each other. The baseline detector has an AUPR approximately equal to the precision (Saito & Rehmsmeier, 2015), and a "perfect" classifier has an AUPR of $100\%$. Consequently, the base rate of the positive class greatly influences the AUPR, so for detection we must specify which class is positive. In view of this, we show the AUPRs when we treat success/normal classes as positive, and then we show the areas when we treat the error/abnormal classes as positive. We can treat the error/abnormal classes as positive by multiplying the scores by $-1$ and labeling them positive. Note that treating error/abnormal classes as positive classes does not change the AU-

---

[1] We consider adversarial example detection techniques in a separate work (Hendrycks & Gimpel, 2016a).

[2] A debatable, imprecise interpretation of AUROC values may be as follows: 90%—100%: Excellent, 80%—90%: Good, 70%—80%: Fair, 60%—70%: Poor, 50%—60%: Fail.

ROC since if $S$ is a score for a successfully classified value, and $E$ is the score for an erroneously classified value, AUROC $= P(S > E) = P(-E > -S)$.

We begin our experiments in Section 3 where we describe a simple baseline which uses the maximum probability from the softmax label distribution in neural network classifiers. Then in Section 4 we describe a method that uses an additional, auxiliary model component trained to reconstruct the input.

## 3 Softmax Prediction Probability as a Baseline

In what follows we retrieve the maximum/predicted class probability from a softmax distribution and thereby detect whether an example is erroneously classified or out-of-distribution. Specifically, we separate correctly and incorrectly classified *test set* examples and, for each example, compute the softmax probability of the predicted class, i.e., the maximum softmax probability.[3] From these two groups we obtain the area under PR and ROC curves. These areas summarize the performance of a binary classifier discriminating with values/scores (in this case, maximum probabilities from the softmaxes) across different thresholds. This description treats correctly classified examples as the positive class, denoted "Success" or "Succ" in our tables. In "Error" or "Err" we treat the the incorrectly classified examples as the positive class; to do this we label incorrectly classified examples as positive and take the negatives of the softmax probabilities of the predicted classes as the scores.

For "In," we treat the in-distribution, correctly classified test set examples as positive and use the softmax probability for the predicted class as a score, while for "Out" we treat the out-of-distribution examples as positive and use the negative of the aforementioned probability. Since the AUPRs for Success, Error, In, Out classifiers depend on the rate of positive examples, we list what area a random detector would achieve with "Base" values. Also in the upcoming results we list the mean predicted class probability of wrongly classified examples (Pred Prob Wrong (mean)) to demonstrate that the softmax prediction probability is a misleading confidence proxy when viewed in isolation. The "Pred. Prob (mean)" columns show this same shortcoming but for out-of-distribution examples.

Table labels aside, we begin experimentation with datasets from vision then consider tasks in natural language processing and automatic speech recognition. In all of the following experiments, the AUROCs differ from the random baselines with high statistical significance according to the Wilcoxon rank-sum test.

### 3.1 Computer Vision

In the following computer vision tasks, we use three datasets: MNIST, CIFAR-10, and CIFAR-100 (Krizhevsky, 2009). MNIST is a dataset of handwritten digits, consisting of 60000 training and 10000 testing examples. Meanwhile, CIFAR-10 has colored images belonging to 10 different classes, with 50000 training and 10000 testing examples. CIFAR-100 is more difficult, as it has 100 different classes with 50000 training and 10000 testing examples.

In Table 1, we see that correctly classified and incorrectly classified examples are sufficiently distinct and thus allow reliable discrimination. Note that the area under the curves degrade with image recognizer test error.

Next, let us consider using softmax distributions to determine whether an example is in- or out-of-distribution. We use all test set examples as the in-distribution (positive) examples. For out-of-distribution (negative) examples, we use realistic images and noise. For CIFAR-10 and CIFAR-100, we use realistic images from the Scene UNderstanding dataset (SUN), which consists of 397 different scenes (Xiao et al., 2010). For MNIST, we use grayscale realistic images from three sources. Omniglot (Lake et al., 2015) images are handwritten characters rather than the handwritten digits in MNIST. Next, notMNIST (Bulatov, 2011) consists of typeface characters. Last of the realistic images, CIFAR-10bw are black and white rescaled CIFAR-10 images. The synthetic "Gaussian" data

---

[3] We also tried using the KL divergence of the softmax distribution from the uniform distribution for detection. With divergence values, detector AUROCs and AUPRs were highly correlated with AUROCs and AUPRs from a detector using the maximum softmax probability. This divergence is similar to entropy (Steinhardt & Liang, 2016; Williams & Renals, 1997).

| Dataset | AUROC /Base | AUPR Succ/Base | AUPR Err/Base | Pred. Prob Wrong(mean) | Test Set Error |
|---|---|---|---|---|---|
| **MNIST** | 97/50 | 100/98 | 48/1.7 | 86 | 1.69 |
| **CIFAR-10** | 93/50 | 100/95 | 43/5 | 80 | 4.96 |
| **CIFAR-100** | 87/50 | 96/79 | 62/21 | 66 | 20.7 |

Table 1: The softmax predicted class probability allows for discrimination between correctly and incorrectly classified test set examples. "Pred. Prob Wrong(mean)" is the mean softmax probability for wrongly classified examples, showcasing its shortcoming as a direct measure of confidence. Succ/Err Base values are the AUROCs or AUPRs achieved by random classifiers. All entries are percentages.

| In-Distribution / Out-of-Distribution | AUROC /Base | AUPR In /Base | AUPR Out/Base | Pred. Prob (mean) |
|---|---|---|---|---|
| **CIFAR-10/SUN** | 95/50 | 89/33 | 97/67 | 72 |
| **CIFAR-10/Gaussian** | 97/50 | 98/49 | 95/51 | 77 |
| **CIFAR-10/All** | 96/50 | 88/24 | 98/76 | 74 |
| **CIFAR-100/SUN** | 91/50 | 83/27 | 96/73 | 56 |
| **CIFAR-100/Gaussian** | 88/50 | 92/43 | 80/57 | 77 |
| **CIFAR-100/All** | 90/50 | 81/21 | 96/79 | 63 |
| **MNIST/Omniglot** | 96/50 | 97/52 | 96/48 | 86 |
| **MNIST/notMNIST** | 85/50 | 86/50 | 88/50 | 92 |
| **MNIST/CIFAR-10bw** | 95/50 | 95/50 | 95/50 | 87 |
| **MNIST/Gaussian** | 90/50 | 90/50 | 91/50 | 91 |
| **MNIST/Uniform** | 99/50 | 99/50 | 98/50 | 83 |
| **MNIST/All** | 91/50 | 76/20 | 98/80 | 89 |

Table 2: Distinguishing in- and out-of-distribution test set data for image classification. CIFAR-10/All is the same as CIFAR-10/(SUN, Gaussian). All values are percentages.

is random normal noise, and "Uniform" data is random uniform noise. Images are resized when necessary.

The results are shown in Table 2. Notice that the mean predicted/maximum class probabilities (Pred. Prob (mean)) are above 75%, but if the prediction probability alone is translated to confidence, the softmax distribution should be more uniform for CIFAR-100. This again shows softmax probabilities should not be viewed as a direct representation of confidence. Fortunately, out-of-distribution examples sufficiently differ in the prediction probabilities from in-distribution examples, allowing for successful detection and generally high area under PR and ROC curves.

For reproducibility, let us specify the model architectures. The MNIST classifier is a three-layer, 256 neuron-wide, fully-connected network trained for 30 epochs with Adam (Kingma & Ba, 2015). It uses a GELU nonlinearity (Hendrycks & Gimpel, 2016b), $x\Phi(x)$, where $\Phi(x)$ is the CDF of the standard normal distribution. We initialize our weights according to (Hendrycks & Gimpel, 2016c), as it is suited for arbitrary nonlinearities. For CIFAR-10 and CIFAR-100, we train a 40-4 wide residual network (Zagoruyko & Komodakis, 2016) for 50 epochs with stochastic gradient descent using restarts (Loshchilov & Hutter, 2016), the GELU nonlinearity, and standard mirroring and cropping data augmentation.

## 3.2 NATURAL LANGUAGE PROCESSING

Let us turn to a variety of tasks and architectures used in natural language processing.

### 3.2.1 SENTIMENT CLASSIFICATION

The first NLP task is binary sentiment classification using the IMDB dataset (Maas et al., 2011), a dataset of polarized movie reviews with 25000 training and 25000 test reviews. This task allows us to determine if classifiers trained on a relatively small dataset still produce informative softmax

| Dataset | AUROC /Base | AUPR Succ/Base | AUPR Err/Base | Pred. Prob Wrong(mean) | Test Set Error |
|---|---|---|---|---|---|
| **IMDB** | 82/50 | 97/88 | 36/12 | 74 | 11.9 |

Table 3: Detecting correct and incorrect classifications for binary sentiment classification.

| In-Distribution / Out-of-Distribution | AUROC /Base | AUPR In /Base | AUPR Out/Base | Pred. Prob (mean) |
|---|---|---|---|---|
| **IMDB/Customer Reviews** | 95/50 | 99/89 | 60/11 | 62 |
| **IMDB/Movie Reviews** | 94/50 | 98/72 | 80/28 | 63 |
| **IMDB/All** | 94/50 | 97/66 | 84/34 | 63 |

Table 4: Distinguishing in- and out-of-distribution test set data for binary sentiment classification. IMDB/All is the same as IMDB/(Customer Reviews, Movie Reviews). All values are percentages.

distributions. For this task we use a linear classifier taking as input the average of trainable, randomly initialized word vectors with dimension 50 (Joulin et al., 2016; Iyyer et al., 2015). We train for 15 epochs with Adam and early stopping based upon 5000 held-out training reviews. Again, Table 3 shows that the softmax distributions differ between correctly and incorrectly classified examples, so prediction probabilities allow us to detect reliably which examples are right and wrong.

Now we use the Customer Review (Hu & Liu, 2004) and Movie Review (Pang et al., 2002) datasets as out-of-distribution examples. The Customer Review dataset has reviews of products rather than only movies, and the Movie Review dataset has snippets from professional movie reviewers rather than full-length amateur reviews. We leave all test set examples from IMDB as in-distribution examples, and out-of-distribution examples are the 500 or 1000 test reviews from Customer Review and Movie Review datasets, respectively. Table 4 displays detection results, showing a similar story to Table 2.

### 3.2.2 TEXT CATEGORIZATION

We turn to text categorization tasks to determine whether softmax distributions are useful for detecting similar but out-of-distribution examples. In the following text categorization tasks, we train classifiers to predict the subject of the text they are processing. In the 20 Newsgroups dataset (Lang, 1995), there are 20 different newsgroup subjects with a total of 20000 documents for the whole dataset. The Reuters 8 (Lewis et al., 2004) dataset has eight different news subjects with nearly 8000 stories in total. The Reuters 52 dataset has 52 news subjects with slightly over 9000 news stories; this dataset can have as few as three stories for a single subject.

For the 20 Newsgroups dataset we train a linear classifier on 30-dimensional word vectors for 20 epochs. Meanwhile, Reuters 8 and Retuers 52 use one-layer neural networks with a bag-of-words input and a GELU nonlinearity, all optimized with Adam for 5 epochs. We train on a subset of subjects, leaving out 5 newsgroup subjects from 20 Newsgroups, 2 news subjects from Reuters 8, and 12 news subjects from Reuters 52, leaving the rest as out-of-distribution examples. Table 5 shows that with these datasets and architectures, we can detect errors dependably, and Table 6 informs us that the softmax prediction probabilities allow for detecting out-of-distribution subjects.

| Dataset | AUROC /Base | AUPR Succ/Base | AUPR Err/Base | Pred.Prob Wrong(mean) | Test Set Error |
|---|---|---|---|---|---|
| **15 Newsgroups** | 89/50 | 99/93 | 42/7.3 | 53 | 7.31 |
| **Reuters 6** | 89/50 | 100/98 | 35/2.5 | 77 | 2.53 |
| **Reuters 40** | 91/50 | 99/92 | 45/7.6 | 62 | 7.55 |

Table 5: Detecting correct and incorrect classifications for text categorization.

| In-Distribution / Out-of-Distribution | AUROC /Base | AUPR In/Base | AUPR Out/Base | Pred. Prob (mean) |
|---|---|---|---|---|
| **15/5 Newsgroups** | 75/50 | 92/84 | 45/16 | 65 |
| **Reuters6/Reuters2** | 92/50 | 100/95 | 56/4.5 | 72 |
| **Reuters40/Reuters12** | 95/50 | 100/93 | 60/7.2 | 47 |

Table 6: Distinguishing in- and out-of-distribution test set data for text categorization.

| Dataset | AUROC /Base | AUPR Succ/Base | AUPR Err/Base | Pred. Prob Wrong(mean) | Test Set Error |
|---|---|---|---|---|---|
| **WSJ** | 96/50 | 100/96 | 51/3.7 | 71 | 3.68 |
| **Twitter** | 89/50 | 98/87 | 53/13 | 69 | 12.59 |

Table 7: Detecting correct and incorrect classifications for part-of-speech tagging.

### 3.2.3 PART-OF-SPEECH TAGGING

Part-of-speech (POS) tagging of newswire and social media text is our next challenge. We use the Wall Street Journal portion of the Penn Treebank (Marcus et al., 1993) which contains 45 distinct POS tags. For social media, we use POS-annotated tweets (Gimpel et al., 2011; Owoputi et al., 2013) which contain 25 tags. For the WSJ tagger, we train a bidirectional long short-term memory recurrent neural network (Hochreiter & Schmidhuber, 1997) with three layers, 128 neurons per layer, with randomly initialized word vectors, and this is trained on $90\%$ of the corpus for 10 epochs with stochastic gradient descent with a batch size of 32. The tweet tagger is simpler, as it is two-layer neural network with a GELU nonlinearity, a weight initialization according to (Hendrycks & Gimpel, 2016c), pretrained word vectors trained on a corpus of 56 million tweets (Owoputi et al., 2013), and a hidden layer size of 256, all while training on 1000 tweets for 30 epochs with Adam and early stopping with 327 validation tweets. Error detection results are in Table 7. For out-of-distribution detection, we use the WSJ tagger on the tweets as well as weblog data from the English Web Treebank (Bies et al., 2012). The results are shown in Table 8. Since the weblog data is closer in style to newswire than are the tweets, it is harder to detect whether a weblog sentence is out-of-distribution than a tweet. Indeed, since POS tagging is done at the word-level, we are detecting whether each word is out-of-distribution given the word and contextual features. With this in mind, we see that it is easier to detect words as out-of-distribution if they are from tweets than from blogs.

| In-Distribution / Out-of-Distribution | AUROC /Base | AUPR In/Base | AUPR Out/Base | Pred. Prob (mean) |
|---|---|---|---|---|
| **WSJ/Twitter** | 80/50 | 98/92 | 41/7.7 | 81 |
| **WSJ/Weblog*** | 61/50 | 88/86 | 30/14 | 93 |

Table 8: Detecting out-of-distribution tweets and blog articles for part-of-speech tagging. All values are percentages. *These examples are atypically close to the training distribution.

### 3.3 AUTOMATIC SPEECH RECOGNITION

Now we consider a task which uses softmax values to construct entire sequences rather than determine an input's class. Our sequence prediction system uses a bidirectional LSTM with two-layers and a clipped GELU nonlinearity, optimized for 60 epochs with RMSProp trained on $80\%$ of the TIMIT corpus (Garofolo et al., 1993). The LSTM is trained with connectionist temporal classification (CTC) (Graves et al., 2006) for predicting sequences of phones given MFCCs, energy, and first and second deltas of a 25ms frame. When trained with CTC, the LSTM learns to have its phone label probabilities spike momentarily while mostly predicting blank symbols otherwise. In this way, the softmax is used differently from typical classification problems, providing a unique test for our detection methods.

We do not show how the system performs on correctness/incorrectness detection because errors are not binary and instead lie along a range of edit distances. However, we can perform out-of-

| In-Distribution / Out-of-Distribution | AUROC /Base | AUPR In/Base | AUPR Out/Base | Pred. Prob (mean) |
|---|---|---|---|---|
| **TIMIT/TIMIT+Airport** | 99/50 | 99/50 | 99/50 | 59 |
| **TIMIT/TIMIT+Babble** | 100/50 | 100/50 | 100/50 | 55 |
| **TIMIT/TIMIT+Car** | 98/50 | 98/50 | 98/50 | 59 |
| **TIMIT/TIMIT+Exhibition** | 100/50 | 100/50 | 100/50 | 57 |
| **TIMIT/TIMIT+Restaurant** | 98/50 | 98/50 | 98/50 | 60 |
| **TIMIT/TIMIT+Street** | 100/50 | 100/50 | 100/50 | 52 |
| **TIMIT/TIMIT+Subway** | 100/50 | 100/50 | 100/50 | 56 |
| **TIMIT/TIMIT+Train** | 100/50 | 100/50 | 100/50 | 58 |
| **TIMIT/Chinese** | 85/50 | 80/34 | 90/66 | 64 |
| **TIMIT/All** | 97/50 | 79/10 | 100/90 | 58 |

Table 9: Detecting out-of-distribution distorted speech. All values are percentages.

distribution detection. Mixing the TIMIT audio with realistic noises from the Aurora-2 dataset (Hirsch & Pearce, 2000), we keep the TIMIT audio volume at 100% and noise volume at 30%, giving a mean SNR of approximately 5. Speakers are still clearly audible to the human ear but confuse the phone recognizer because the prediction edit distance more than doubles. For more out-of-distribution examples, we use the test examples from the THCHS-30 dataset (Wang & Zhang, 2015), a Chinese speech corpus. Table 9 shows the results. Crucially, when performing detection, we compute the softmax probabilities while ignoring the blank symbol's logit. With the blank symbol's presence, the softmax distributions at most time steps predict a blank symbol with high confidence, but without the blank symbol we can better differentiate between normal and abnormal distributions. With this modification, the softmax prediction probabilities allow us to detect whether an example is out-of-distribution.

# 4 ABNORMALITY DETECTION WITH AUXILIARY DECODERS

Having seen that softmax prediction probabilities enable abnormality detection, we now show there is other information sometimes more useful for detection. To demonstrate this, we exploit the learned internal representations of neural networks. We start by training a normal classifier and append an auxiliary decoder which reconstructs the input, shown in Figure 1. Auxiliary decoders are sometimes known to increase classification performance (Zhang et al., 2016). The decoder and scorer are trained jointly on in-distribution examples. Thereafter, the blue layers in Figure 1 are frozen. Then we train red layers on clean and noised training examples, and the sigmoid output of the red layers scores how normal the input is. Consequently, noised examples are in the abnormal class, clean examples are of the normal class, and the sigmoid is trained to output to which class an input belongs. After training we consequently have a normal classifier, an auxiliary decoder, and what we call an **abnormality module**. The gains from the abnormality module demonstrate there are possible research avenues for outperforming the baseline.

## 4.1 TIMIT

We test the abnormality module by revisiting the TIMIT task with a different architecture and show how these auxiliary components can greatly improve detection. The system is a three-layer, 1024-neuron wide classifier with an auxiliary decoder and abnormality module. This network takes as input 11 frames and must predict the phone of the center frame, 26 features per frame. Weights are initialized according to (Hendrycks & Gimpel, 2016c). This network trains for 20 epochs, and the abnormality module trains for two. The abnormality module sees clean examples and, as negative examples, TIMIT examples distorted with either white noise, brown noise (noise with its spectral density proportional to $1/f^2$), or pink noise (noise with its spectral density proportional to $1/f$) at various volumes.

We note that the abnormality module is *not* trained on the same type of noise added to the test examples. Nonetheless, Table 10 shows that simple noised examples translate to effective detection of realistically distorted audio. We detect abnormal examples by comparing the typical abnormality

| In-Distribution / Out-of-Distribution | AUROC /Base Softmax | AUROC /Base AbMod | AUPR In/Base Softmax | AUPR In/Base AbMod | AUPR Out/Base Softmax | AUPR Out/Base AbMod |
|---|---|---|---|---|---|---|
| **TIMIT/+Airport** | 75/50 | 100/50 | 77/41 | 100/41 | 73/59 | 100/59 |
| **TIMIT/+Babble** | 94/50 | 100/50 | 95/41 | 100/41 | 91/59 | 100/59 |
| **TIMIT/+Car** | 70/50 | 98/50 | 69/41 | 98/41 | 70/59 | 98/59 |
| **TIMIT/+Exhib.** | 91/50 | 98/50 | 92/41 | 98/41 | 91/59 | 98/59 |
| **TIMIT/+Rest.** | 68/50 | 95/50 | 70/41 | 96/41 | 67/59 | 95/59 |
| **TIMIT/+Subway** | 76/50 | 96/50 | 77/41 | 96/41 | 74/59 | 96/59 |
| **TIMIT/+Street** | 89/50 | 98/50 | 91/41 | 99/41 | 85/59 | 98/59 |
| **TIMIT/+Train** | 80/50 | 100/50 | 82/41 | 100/41 | 77/59 | 100/59 |
| **TIMIT/Chinese** | 79/50 | 90/50 | 41/12 | 66/12 | 96/88 | 98/88 |
| Average | 80 | 97 | 77 | 95 | 80 | 98 |

Table 10: Abnormality modules can generalize to novel distortions and detect out-of-distribution examples even when they do not severely degrade accuracy. All values are percentages.

| In-Distribution / Out-of-Distribution | AUROC /Base Softmax | AUROC /Base AbMod | AUPR In/Base Softmax | AUPR In/Base AbMod | AUPR Out/Base Softmax | AUPR Out/Base AbMod |
|---|---|---|---|---|---|---|
| **MNIST/Omniglot** | 95/50 | 100/50 | 95/52 | 100/52 | 95/48 | 100/48 |
| **MNIST/notMNIST** | 87/50 | 100/50 | 88/50 | 100/50 | 90/50 | 100/50 |
| **MNIST/CIFAR-10bw** | 98/50 | 100/50 | 98/50 | 100/50 | 98/50 | 100/50 |
| **MNIST/Gaussian** | 88/50 | 100/50 | 88/50 | 100/50 | 90/50 | 100/50 |
| **MNIST/Uniform** | 99/50 | 100/50 | 99/50 | 100/50 | 99/50 | 100/50 |
| Average | 93 | 100 | 94 | 100 | 94 | 100 |

Table 11: Improved detection using the abnormality module. All values are percentages.

module outputs for clean examples with the outputs for the distorted examples. The noises are from Aurora-2 and are added to TIMIT examples with 30% volume. We also use the THCHS-30 dataset for Chinese speech. Unlike before, we use the THCHS-30 training examples rather than test set examples because fully connected networks can evaluate the whole training set sufficiently quickly. It is worth mentioning that *fully connected* deep neural networks are noise robust (Seltzer et al., 2013), yet the abnormality module can still detect whether an example is out-of-distribution. To see why this is remarkable, note that the network's frame classification error is 29.69% on the *entire* test (not core) dataset, and the average classification error for distorted examples is 30.43%—this is unlike the bidirectional LSTM which had a more pronounced performance decline. Because the classification degradation was only slight, the softmax statistics alone did not provide useful out-of-distribution detection. In contrast, the abnormality module provided scores which allowed the detection of different-but-similar examples. In practice, it may be important to determine whether an example is out-of-distribution even if it does not greatly confuse the network, and the abnormality module facilitates this.

## 4.2 MNIST

Finally, much like in a previous experiment, we train an MNIST classifier with three layers of width 256. This time, we also use an auxiliary decoder and abnormality module rather than relying on only softmax statistics. For abnormal examples we blur, rotate, or add Gaussian noise to training images. Gains from the abnormality module are shown in Table 11, and there is a consistent out-of-sample detection improvement compared to softmax prediction probabilities. Even for highly dissimilar examples the abnormality module can further improve detection.

## 5 DISCUSSION AND FUTURE WORK

The abnormality module demonstrates that in some cases the baseline can be beaten by exploiting the representations of a network, suggesting myriad research directions. Some promising future avenues may utilize the intra-class variance: if the distance from an example to another of the same predicted class is abnormally high, it may be out-of-distribution (Giryes et al., 2015). Another path is to feed in a vector summarizing a layer's activations into an RNN, one vector for each layer. The RNN may determine that the activation patterns are abnormal for out-of-distribution examples. Others could make the detections fine-grained: is the out-of-distribution example a known-unknown or an unknown-unknown? A different avenue is not just to detect correct classifications but to output the probability of a correct detection. In Appendix B, we show a baseline and evaluation metrics that future research can utilize for estimating the probability of a correct classification. These are but a few ideas for improving error and out-of-distribution detection.

We hope that any new detection methods are tested on a variety of tasks and architectures of the researcher's choice. A basic demonstration could include the following datasets: MNIST, CIFAR, IMDB, and tweets because vision-only demonstrations may not transfer well to other architectures and datasets. Reporting the AUPR and AUROC values is important, and so is the underlying classifier's accuracy since an always-wrong classifier gets a maximum AUPR for error detection if error is the positive class. Also, future research need not use the exact values from this paper for comparisons. Machine learning systems evolve, so tethering the evaluations to the exact architectures and datasets in this paper is needless. Instead, one could simply choose a variety of datasets and architectures possibly like those above and compare their detection method with a detector based on the softmax prediction probabilities from their classifiers. These are our basic recommendations for others who try to surpass the baseline on this underexplored challenge.

## 6 CONCLUSION

We demonstrated a softmax prediction probability baseline for error and out-of-distribution detection across several architectures and numerous datasets. We then presented the abnormality module, which provided superior scores for discriminating between normal and abnormal examples on tested cases. The abnormality module demonstrates that the baseline can be beaten in some cases, and this implies there is room for future research. Our hope is that other researchers investigate architectures which make predictions in view of abnormality estimates, and that others pursue more reliable methods for detecting errors and out-of-distribution inputs because knowing when a machine learning system fails strikes us as highly important.

### ACKNOWLEDGMENTS

We would like to thank John Wieting, Hao Tang, Karen Livescu, Greg Shakhnarovich, and our reviewers for their suggestions. We would also like to thank NVIDIA Corporation for donating several TITAN X GPUs used in this research.

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

# A   ABNORMALITY MODULE EXAMPLE

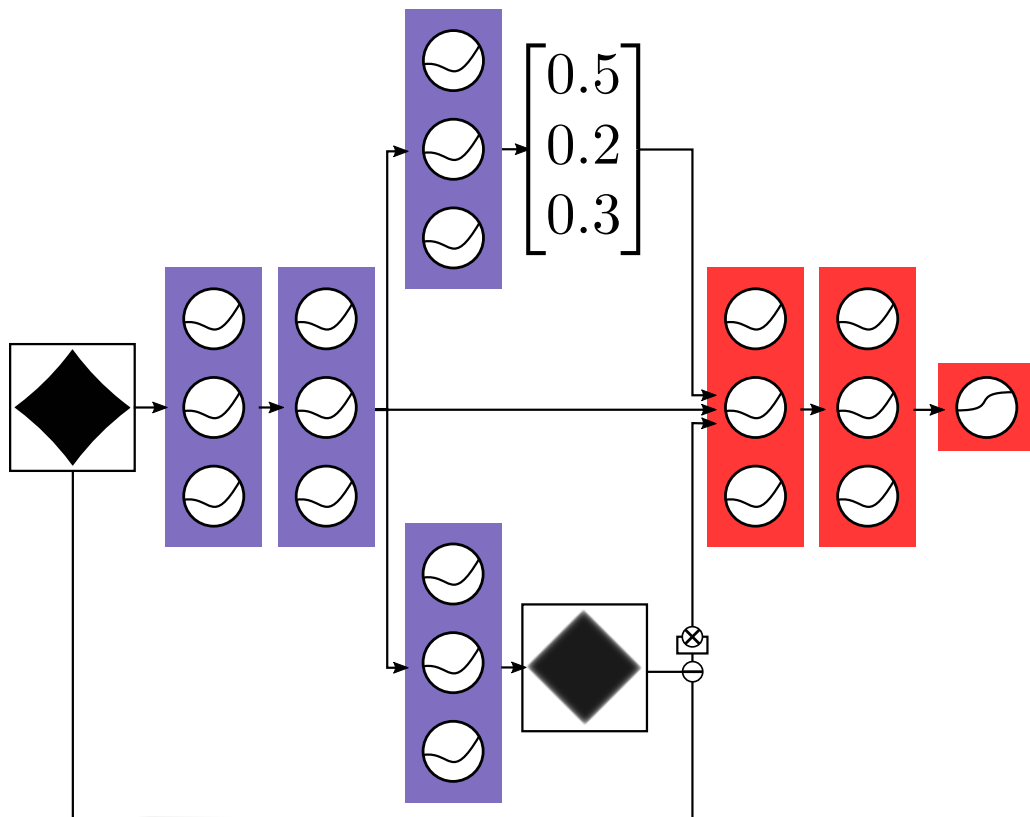

Figure 1: A neural network classifying a diamond image with an auxiliary decoder and an abnormality module. Circles are neurons, either having a GELU or sigmoid activation. The blurred diamond reconstruction precedes subtraction and elementwise squaring. The probability vector is the softmax probability vector. Blue layers train on in-distribution data, and red layers train on both in- and out-of-distribution examples.

## B  Metrics for Meaningful Confidence Values

Throughout this work, we show that the softmax probability of a predicted class is consistently high and thus is a poor proxy for estimating the "confidence" in a classification. In addition to being frequently high, the softmax prediction probability is bounded below by a value greater than zero. To see this, note that a binary sentiment classifier's minimum confidence in an out-of-distribution example is 50% since the highest softmax probability must be at least 50%. Yet for an out-of-distribution example we may want less confidence if confidence is to reflect the probability of correct classification.

To evaluate how well a confidence model reflects the probability of correct classification, we present two metrics. First, let us establish notation. Denote the confidence model by $c : \mathcal{X} \to [0, 1]$, where $\mathcal{X}$ is a set of inputs. An output near 1 from $c$ reflects high confidence, or a high probability that the underlying classification model $f : \mathcal{X} \to \mathcal{Y}$ correctly classified its input. Also, let $\mathbb{1}(b)$ be 1 if $b$ is True, and 0 if $b$ is False. A possible confidence model evaluation metric is accuracy. To compute this, we must threshold the confidence at 0.5. However, the model can output probabilities slightly above or below 0.5 and obtain perfect accuracy. Probabilities all near 0.5 are insufficiently informative and lack numerical resolution, and for this reason we avoid thresholding in our metrics. Instead, we use the confidence itself in a metric which we call the *probability alignment* score,

$$\mathbb{E}_{(x,y)\sim(\mathcal{X},\mathcal{Y})}[2\mathbb{1}(f(x) = y) - 1][2c(x) - 1].$$

Note that if $f$ is a highly accurate model, then the confidence model can obtain a high probability alignment by predicting 1 constantly. We meet this challenge by also calculating a "soft" F1 score (Pastor-Pellicer et al., 1998) for a test set $\{(x_i, y_i)\}_{i=1}^m$. We define the soft F1 score by first letting $\text{tp} = \sum_{i=1}^m (1 - c(x_i))\mathbb{1}(f(x_i) \neq y_i)$, $\text{fp} = \sum_{i=1}^m (1 - c(x_i))\mathbb{1}(f(x_i) = y_i)$, $\text{fn} = \sum_{i=1}^m c(x_i)\mathbb{1}(f(x_i) \neq y_i)$, $\text{pr} = \text{tp}/(\text{tp} + \text{fp})$, and $\text{rc} = \text{tp}/(\text{tp} + \text{fn})$. Then the soft F1 score is

$$2\frac{\text{pr} \cdot \text{rc}}{\text{pr} + \text{rc}},$$

and this is our second and final metric for evaluating a confidence model.

Using these metrics, we now establish a baseline using the logits of a softmax. Let $l_i$ represent the logits vector (the input to the softmax) for example $x_i$. Our baseline confidence model is $c(x_i) = \sigma(\max l_i)$, where $\sigma$ is the logistic sigmoid. Let us evaluate this baseline by using the same MNIST, Twitter, and IMDB setups from before. We take the median result over five runs. In the case of CIFAR-10, we train using a 40-2 WideResNet. The results are in Table 12, and they indicate that the probabilities are positively aligned and have a nonzero soft F1 score. However, these results also indicate wide room for improvement.

| Dataset | Probability Alignment Score | Soft F1 Score | Test Set Error |
|---|---|---|---|
| **MNIST** | 97 | 24 | 1.56 |
| **CIFAR-10** | 88 | 28 | 5.42 |
| **Twitter** | 74 | 29 | 13.6 |
| **IMDB** | 67 | 26 | 13.4 |

Table 12: Confidence Model Baseline Results. All values are percentages.

In this appendix, we presented a simple baseline method for estimating classification confidence and metrics for evaluating the confidence model. The baseline draws on the idea of using softmax classification information to create a confidence model. We hope this appendix gives ideas for future research directions, serves as a baseline, or provides metrics for confidence model evaluation.

