# Peer review of "A Baseline for Detecting Misclassified and Out-of-Distribution Examples in Neural Networks"

_ICLR 2017 — accepted_

[Official Review · AnonReviewer2 · rating 6 · confidence 3 · 16 Dec 2016]
**Paper provides a simple baseline for out-of-domain/misclassification detection. Statistics on maximum softmax probabilities for in/out domain examples appear to be sufficient to classify examples as out-of-domain.**

The authors present results on a number of different tasks where the goal is to determine whether a given test example is out-of-domain or likely to be mis-classified. This is accomplished by examining statistics for the softmax probability for the most likely class; although the score by itself is not a particularly good measure of confidence, the statistics for out-of-domain examples are different enough from in-domain examples to allow these to be identified with some certainty. 

My comments appear below:
1. As the authors point out, the AUROC/AUPR criterion is threshold independent. As a result, it is not obvious whether the thresholds that would correspond to a certain operating point (say a true positive rate of 10%) would be similar across different data sets. In other words, it would be interesting to know how sensitive the thresholds are to different test sets (or different splits of the test set). This is important if we want to use the thresholds determined on a given held-out set during evaluation on unseen data (where we would need to select a threshold).

2. Performance is reported in terms of AUROC/AUPR and models are compared against a random baseline. I think it’s a little hard to look at the differences in AUC/AUPR to get a sense for how much better the proposed classifier is than the random baseline. It would be useful, for example, if the authors could also report how strongly statistically significant some of these differences are (although admittedly they look to be pretty large in most cases).

3. In the experiments on speech recognition presented in Section 3.3, I was not entirely clear on how the model was evaluated. In Table 9, for example, is an “example” the entire utterance or just a single (stacked?) speech frame. Assuming that each “example” is an utterance, are the softmax probabilities the probability of the entire phone sequence (obtained by multiplying the local probability estimates from a Viterbi decoding?)

4. I’m curious about the decision to ignore the blank symbol’s logit in Section 3.3. Why is this required?

5. As I mentioned in the pre-review question, at least in the speech recognition case, it would have been interesting to compare performance obtained using a simple generative baseline (e.g., GMM-HMM). I think that would serve as a good indication of the ability of the proposed model to detect out-of-domain examples over the baseline.

[Official Review · AnonReviewer1 · rating 6 · confidence 3 · 19 Dec 2016]
**Important topic**
originality 4 · meaningful comparison 4

The paper address the problem of detecting if an example is misclassified or out-of-distribution. This is an very important topic and the study provides a good baseline. Although it misses strong novel methods for the task, the study contributes to the community.

[Official Review · AnonReviewer3 · rating 6 · confidence 3 · 19 Dec 2016 (modified: 20 Dec 2016)]
**Paper explores the problem of classifier accuracy estimation and out of domain probability estimation.**
soundness 3 · originality 3 · impact 2

The authors propose the use of statistics of softmax outputs to estimate the probability of error and probability of a test sample being out-of-domain. They contrast the performance of the proposed method with directly using the softmax output probabilities, and not their statistics, as a measure of confidence.

It would be great if the authors elaborate on the idea of ignoring the logit of the blank symbol.

It would be interesting to see the performance of the proposed methods in more confusable settings, ie., in cases where the out-of-domain examples are more similar to the in-domain examples. e.g., in the case of speech recognition this might correspond to using a different language's speech with an ASR system trained in a particular language. Here the acoustic characteristics of the speech signals from two different languages might be more similar as compared to the noisy and clean speech signals from the same language.

In section 4, the description of the auxiliary decoder setup might benefit from more detail.

There has been recent work on performance monitoring and accuracy prediction in the area of speech recognition, some of this work is listed below. 
1. Ogawa, Tetsuji, et al. "Delta-M measure for accuracy prediction and its application to multi-stream based unsupervised adaptation." Proceedings of ICASSP. 2015.

2. Hermansky, Hynek, et al. "Towards machines that know when they do not know." Proceedings of ICASSP, 2015.

3. Variani, Ehsan et al. "Multi-stream recognition of noisy speech with performance monitoring." INTERSPEECH. 2013.

[Author Response · Dan Hendrycks · 14 Jan 2017]
**Paper Update**

We have updated the paper to make mention of the statistical significance of the AUROCs. We also added Chinese speech from THCHS-30 as out-of-distribution examples. Finally, we added more information about the logit of the blank symbol, the training of the abnormality module, and we revised Appendix B.

[Final Decision · Program Chairs · 06 Feb 2017]
**ICLR committee final decision**

The paper presents an approach that uses the statistics of softmax outputs to identify misclassifications and/or outliers. The reviewers had mostly minor comments on the paper, which appear to have been appropriately addressed in the revised version of the paper.